# Double-Lumen Endotracheal Tube—Predicting Insertion Depth and Tube Size Based on Patient’s Chest X-ray Image Data and 4 Other Body Parameters

**DOI:** 10.3390/diagnostics12123162

**Published:** 2022-12-14

**Authors:** Tsai-Rong Chang, Mei-Kang Yuan, Shao-Fang Pan, Chia-Chun Chuang, Edmund Cheung So

**Affiliations:** 1Department of Computer Science and Information Engineering, Southern Taiwan University of Science and Technology, Tainan 71005, Taiwan; 2Department of Radiology, An-Nan Hospital, China Medical University, Tainan 70965, Taiwan; 3Department of Medical Imaging and Radiology, Shu-Zen Junior College of Medicine and Management, Kaohsiung 82144, Taiwan; 4Department of Cell Biology and Anatomy College of Medicine, National Cheng Kung University, Tainan 70101, Taiwan; 5Department of Anesthesiology, An-Nan Hospital, China Medical University, Tainan 70965, Taiwan

**Keywords:** double lumen endotracheal tube, medical imaging, machine learning, predictive modeling, support vector machine

## Abstract

In thoracic surgery, the double lumen endotracheal tube (DLT) is used for differential ventilation of the lung. DLT allows lung collapse on the surgical side that requires access to the thoracic and mediastinal areas. DLT placement for a given patient depends on two settings: a tube of the correct size (or ‘size’) and to the correct insertion depth (or ‘depth’). Incorrect DLT placements cause oxygen desaturation or carbon dioxide retention in the patient, with possible surgical failure. No guideline on these settings is currently available for anesthesiologists, except for the aid by bronchoscopy. In this study, we aimed to predict DLT ‘depths’ and ‘sizes’ applied earlier on a group of patients (*n* = 231) using a computer modeling approach. First, for these patients we retrospectively determined the correlation coefficient (r) of each of the 17 body parameters against ‘depth’ and ‘size’. Those parameters having r > 0.5 and that could be easily obtained or measured were selected. They were, for both DLT settings: (a) sex, (b) height, (c) tracheal diameter (measured from X-ray), and (d) weight. For ‘size’, a fifth parameter, (e) chest circumference was added. Based on these four or five parameters, we modeled the clinical DLT settings using a Support Vector Machine (SVM). After excluding statistical outliers (±2 SD), 83.5% of the subjects were left for ‘depth’ in the modeling, and similarly 85.3% for ‘size’. SVM predicted ‘depths’ matched with their clinical values at a r of 0.91, and for ‘sizes’, at an r of 0.82. The less satisfactory result on ‘size’ prediction was likely due to the small target choices (*n* = 4) and the uneven data distribution. Furthermore, SVM outperformed other common models, such as linear regression. In conclusion, this first model for predicting the two DLT key settings gave satisfactory results. Findings would help anesthesiologists in applying DLT procedures more confidently in an evidence-based way.

## 1. Introduction

Since the establishment of artificial respiration, the technique of lung isolation began to develop around the early 1900s, mainly to prevent infected secretions from one side of the lung cross contaminating the other lung [1]. Nowadays, in thoracic surgery, lung isolation has become an essential or standard procedure for most patients. Lung isolation has not only facilitated the protective purpose but has also made one lung ventilation possible. The one lung ventilation technique allows the controlled collapse of the operative lung facilitating surgical access into the thoracic and mediastinal area [2]. Specially designed equipment, such as single lumen endotracheal tubes (SLTs), bronchial blockers (BBs), or double lumen endotracheal tubes (DLTs), can all be used to achieve lung isolation, however DLTs eventually became the most popular choice among anesthesiologists, primarily because they are easier and quicker to place [3].

The concept of a DLT first appeared in 1889 and has been continuously modified over the last century [1]. Modern day DLT designs are based on the design published by Carlens in 1949 [4], which was later refined by Robertshaw [5], and have since been popular in practice until now [6]. DLTs today are single-use, disposable equipment made of polyvinyl chloride (PVC), instead of rubber due to sterilization concerns [2]. They are composed of two endotracheal tubes bonded together: the shorter tracheal lumen ends in the distal trachea when placed correctly, and the longer, angled bronchial lumen enters either the left or right main bronchus based on its sidedness. In addition, they also possess balloon-like tracheal and bronchial cuffs, which can block the according position when inflated, further allowing the anesthesiologist to control which lung they choose to ventilate.

In most clinical situations, left-sided DLTs are preferred over right-sided DLTs, while both designs allow individual ventilation for either lung. Although the angle of the right main bronchus (RMB) is generally more aligned with the trachea, there is also a greater risk of obstructing the ventilation of the right upper lobe when using a right-sided DLT, especially in some patients whose right upper lobar bronchus originated near the carina or even from the trachea [7,8]. Therefore, right-sided DLTs are usually used as a last approach in situations such as when there is an obstruction of the LMB, or when there is a surgical procedure involving the proximal LMB [9]. Since the bifurcation of the left main bronchus (LMB) usually occurs more distal than the RMB, it is considered to have a wider safety margin, which is defined as the length that a DLT may be moved or positioned without obstructing a conducting airway [10].

Incorrect placement of the DLT during surgery can cause many problems, such as airway trauma or rupture; incidents such as accidental blockage of the left upper lobe by the bronchial cuff leading to hypoxemia during one lung ventilation can also occur [11]. The correct placement of a DLT depends a lot on choosing the best fitting DLT size for the patient. Using an undersized DLT can cause airway injuries because of the need to use higher pressures in the bronchial cuff to completely seal off the space between the outer border of the DLT and the airway; the smaller internal diameter of the DLT can also increase airway resistance during mechanical ventilation, which could lead to auto positive end-expiratory pressure (PEEP). DLTs that are too small also have higher chances of dislodgment, which can lead to the failure of lung isolation and an inability to suction secretions [8]. On the other hand, the use of oversized DLTs have been observed to correlate with postoperative sequelae, such as a sore throat and hoarseness. Because larger sized DLTs are more difficult to intubate, this could also lead to forceful insertions and repeated attempts of intubation, causing airway and teeth injury during the process, and the extra wasting of time and equipment [12].

A fundamental problem for choosing an adequately sized DLT for individual patients is the lack of evidence-based guidelines. It is traditionally recommended to select DLT size based on the patient’s gender and height, however this method is not very accurate, especially in females, persons with smaller statures, and Asian populations [2,8]. Using patients’ airway measurements acquired from various imaging techniques, such as chest X-rays (CXR), computed tomography (CT), magnetic resonance imaging (MRI) or ultrasound (UR), to facilitate the choosing of DLT sizes appears to be a more accurate approach [12,13,14,15,16]. However, this method still has its shortcomings, for example, this is not applicable for emergency trauma patients who may not have time to perform chest CT.

In this study, our goal is to analyze tracheobronchial tree measurements from the clinical imaging records of adult patients who received DLT intubation in Tainan Municipal An-Nan Hospital-China Medical University. From the dataset, we have analyzed the correlation between patients’ demographic data (such as age, sex and body sizes), in the hope of developing a formula via machine learning that can be used to predict the best fit DLT size for the local population. The results would allow us to expand the analysis to a larger dataset and develop a mobile formula-based application, which can provide clinicians with easy and actual application in real life.

## 2. Materials and Methods

### 2.1. Patient Population

The following descriptions are based on the dataset. This retrospective, single-center study had already been approved by the Institutional Review Board of Tainan Municipal An-Nan Hospital-China Medical University (TMANH110-REC002). The data of patients who received thoracotomy/thoracoscopy using double lumen endotracheal tube at An-Nan hospital between 2013–2021 were retrieved from the hospital database. Demographic information including age, sex, body weight, height, America Society of Anesthesiologist (ASA) score, tube fixation depth and size of DLT used in surgery were recorded for later analysis.

### 2.2. Double Lumen Endotracheal Tube

The left-type “Covidien” Mallinckrodt Broncho-cath Endobronchial Tube (sizes 28, 32, 35 and 37 Fr) was used in this study (Figure 1).

### 2.3. Chest Computed Tomography and X-ray Measurements

The patient’s roentgenogram was performed within one month before the thoracotomy or thoracoscopy was retrieved from the hospital’s Picture Archiving and Communication System (PACS). All measurements were acquired using the built-in measuring tool in PACS. Since the cricoid ring is the narrowest part of the trachea, its diameter determines whether the DLT can successfully pass through the subglottic region or not. Therefore, the transverse diameter (TD) and the anteroposterior diameter (APD) of the cricoid ring were measured at its lower border, which can be identified as the last slice where the cricoid cartilage is visible (Figure 2).

The bronchial cuff of a left DLT is usually placed in the left main bronchus (LMB) at 1 cm below the carina, therefore the dimensions of the LMB were measured at this point. The TD-LMB was measured on two consecutive coronal slices (Figure 3), then these two measurements were averaged to reduce sampling error. Next, using the cross-link function in PACS, the “1 cm below carina” position on the axial and sagittal slice was located (where the two planes intersect with the midpoint of transverse diameter), and the APD-LMB was measured at this location (Figure 4). The two measurements from the axial and sagittal slices were also averaged to reduce sampling error.

Considering that the elasticity of the bronchus could cause its shape to slightly shift during intubation, the equivalent diameter of the left main bronchus (ED-LMB) was also calculated based on the patient’s TD-LMB and APD-LMB using the perimeter formula of the ellipse ($\pi \left[3\left(a+b\right)-\sqrt{\left(a+3b\right)\left(3a+b\right)}\right],$ where a is the larger and b is the smaller radius), and the perimeter formula of the circle (π · D).

Since we also wanted to investigate whether the patient’s bust correlates with their airway measurements in this study, the TD, APD and circumference of the patient’s chest were measured at the 7th intercostal space (Figure 5 and Figure 6).

Because chest X-rays are performed more commonly than CTs and is a routine procedure for most patients before undergoing thoracic surgery, we also wanted to investigate whether chest X-ray data can be used to predict a patient’s DLT size. Three different chest X-rays taken before the patient’s surgery (which also possess a clear view of the trachea) were chosen and retrieved from PACS to be used in our measurements. On each X-ray, the TD-trachea was measured at the mid-body of C7 using the built-in measuring tool in PACS (Figure 5); the three measurements were averaged to reduce sampling error.

### 2.4. Final Set of 17 Parameters

We finally collected a dataset of 17 parameters (Table 1) for each of the 231 patients.

## 3. Data Processing and Statistical Analysis

The first step was performing a correlational ranking of all the parameters: for all 231 subjects, each of their 17 body parameters were linearly correlated with either ‘size’ or ’depth’ values. Their correlation coefficient, or r values, were then ordered from high-to-low, for ‘depth’ and ‘size’, respectively. The easily available parameters with r values > 0.5 for ‘depth’ and ‘size’ separately were identified. A Support Vector Machine (SVM) was then adopted to model the clinical data of ‘size’ or ‘depth’ based on the four or five body parameters identified. A detailed description of SVM is given separately in Appendix A. In brief, it is a machine learning model using a supervised learning model with associated algorithms, typically applied for classification and regression analyses. Support Vector Machine (SVM) is a supervised algorithm based on statistical learning that separates two distinct sets by a hyperplane. The classification is to find the dividing line between different classes of data. In general, the dividing line is complex with multiple possibilities. SVM is to find the best solution among these possibilities. The spirit of the SVM algorithm is to find a separation line (or hyperplane) so that all points on the boundary are separated as far as possible, to better resist noise in data. It has advantages over the robust linear regression model (details shown in Appendix A) when data distribution is not normal, as in our present study. For comparison purposes, we also tested a linear regression model at a later stage. For example, a cell-phone with a dedicated software would allow clinicians to obtain values on ‘depth’ and ‘size’ almost instantly after entering the five parameters.

The computer language used for modeling was Python [17], which is a general-purpose language with a wide array of modules and tool packages that can be used for statistical analysis. In this study, statistical analyses were conducted in Jupyter Notebook (Version 6.3.0) using Python packages Pandas, NumPy, Mlxtend, Matplotlib and scikit-learn.

## 4. Results

### 4.1. Demographic Data

In this study, data of 294 patients were first collected from the hospital database. After excluding those who were missing one or more roentgenogram measurements, missing intubation records, or did not receive a left-DLT intubation during surgery, a total of 231 patients were included for study.

Of the 231 patients included in this study, 161 were male (69.7%) and 70 were female (30.3%). For the male group, the mean age was 57 ± 16.8 years old, mean weight was 66.2 ± 14.8 kg, mean height was 167.1 ± 6.3 cm, and mean BMI was 23.6 ± 4.8; for the female group, the mean age was 59.5 ± 14.5 years old, mean weight was 57.4 ± 10.2 kg, mean height was 154.7 ± 5.4 cm, and mean BMI was 24.6 ± 4.4; and for the overall population, the mean age was 57.7 ± 15.4 years old, mean weight was 63.5 ± 14.1 kg, mean height was 163.4 ± 8.3 cm, and mean BMI was 23.8 ± 4.7. Table 2 shows details of their statistics.

Regarding intubation data, 9 patients were intubated with a DLT size of 28 Fr (3.9%), 52 with 32 Fr (22.5%), 128 with 35 Fr (55.4%), and 42 with 37 Fr (18.2%) (Table 3). For males, the DLT size mostly used was 35 Fr (104 people, 64.6% of the sex group), and for females, the size was 32 Fr (40 people, 57.1% of the sex group). No female patient was intubated with the 37 Fr DLT. The mean intubation depth for male patients was 29.7 ± 1.1 cm, for female patients it was 27.2 ± 1.1 cm, and overall it was 29.0 ± 1.6 cm. The mean size of the patients’ trachea, cricoid, LMB, and chest, plus the angle between RMB/trachea versus LMB as measured from their roentgenogram, are summarized in Table 4. As expected, the RMB-LMB angle, all the other measurements for females were smaller than males.

### 4.2. Ranking of 17 Body Parameters According to Correlation Coefficient with ‘Depth’ or ‘Size’

The Pearson correlation coefficient (r) was performed for 19 body parameters of all 231 subjects, with either ‘size’ or ’depth’ values. The r values were ranked from high to low, as shown in Table 5. We identified for either the ‘size’ or ‘depth’ dataset, the parameters meeting the following two criteria: (a) r > 0.5, and (b) either easily obtained in the medical records of a patient, or easily measured by a chest X-ray. According to these criteria, we identified four parameters common for both ‘depth’ and ‘size’. They were (from high to low r): (a) sex, (b) body height, (c) tracheal diameter (easily measured from chest X-ray), and (d) body weight. An additional parameter: (e) chest circumference, was also identified for ‘size’ (with r also similar to body weight) (Table 5). Some parameters with r > 0.5 were not included (such as those associated with CRICOID, LMB, or CHEST) because they were measured from CT scans, which were less available than the standard chest X-ray.

### 4.3. Exclusion of Statistical Outliers in Data from the Selected Four or Five Body Parameters

To improve model performance, we excluded statistical outliers (outside ± 2 SD of individual body parameters), according to a similar approach we adopted earlier in predicting the tidal volume of patients based on their body parameters [18]. The resulting distributions are shown in histograms (Figure 7), as well as a scatterplot (Appendix A). The excluded subjects (each having any body parameter associated with the exclusion) were for ‘depth’, 38 subjects (or 16.5% of population) and for ‘size’, 34 subjects (or 14.7% of population). Finally, the number of subjects entering the next stage of modeling was 193 (or 83.5% of population) for ‘depth’ and 197 (or 85.3% of population) for ‘size’ modeling.

### 4.4. Support Vector Machine (SVM) Modeling Results

Based on outlier-rejected data of the four or five selected parameters, the SVM modeling was performed. The predicted values of ‘depth’ and ‘size’ are plotted in Figure 8. The values of ‘depth’ were better correlated with the clinical values at a correlation coefficient r of 0.91, compared with the r of 0.82 for ‘size’. The less satisfactory result on ‘size’ is likely due to the small target choices (*n* = 4) and the uneven data distribution, despite having one more parameter in modeling. In addition, the time for SVM modeling was typically fast (<70 msec) for processing the data from 231 patients.

### 4.5. Modeling Results from Linear Regression Compared with SVM

Prediction results were also obtained with another more common model: linear regression. The performance was less satisfactory compared with SVM (Table 6). To show the effect of individual parameters, results with leave-one-out modeling are also shown.

## 5. Discussion

The principal finding of this study is that the DLT both ‘size’ and ‘depth’ were modeled rather satisfactorily, using the SVM approach. Inputs to the model were four or five body parameters, including one measured from chest X-ray images. All of these parameters are easily available from the medical records of patients. Chest X-ray images are also routinely available. Results fulfilled our primary aim in finding body parameters that could be conveniently used to predict ‘size’ and ‘depth’ settings in the DLT procedure, especially for patients in emergent situations.

Since we had measured other parameters such as those from CT images, and calculated their r values, we performed an alternative modeling based on the top five parameters with r values >0.65. Despite their higher r values, only slightly better results were found (Appendix A). The reason for such minimal improvements is likely due to the high correlation among those new parameters themselves (r values: 0.72 to 0.79, results not shown).

The fact that body parameters are able to predict chest-related metric, is first consistent with our report predicting the tidal volume of the lung based on similar body parameters. For tidal volume, the six parameters used were, in descending order of importance: weight, height, chest circumference, sex, BMI and age. The first four parameters overlap with those in our present study. In the tidal volume study, we applied only linear regression analysis. There, data were more evenly distributed (giving rise to higher values of r) compared with our present study. Given the relatively low values of r of the four or five parameters against ‘depths’ or ‘sizes’, the hyperplane classifier as provided by SVM, is likely more appropriate than linear regression for modeling. Since hyperplane can better handle the characteristics of our data, such as the small number of model targets, and uneven distribution. These, and other data characteristics, might well lead to the discrepancy in model performance between ‘depth’ and ‘size’.

The pre-processing step of rejecting statistical outliers in the dataset was used in the present study. The rationale was to produce a more homogeneous dataset for modeling. Without outlier rejection, model performance was expectedly poorer (Appendix A).

There were similar studies focusing on using airway parameters to predict the best fit DLT size for individual patients [12,14,15,16]. Most of these studies used CT imaging and multiplanar reconstruction (MPR) to measure the cricoid and LMB for subjective determination of the DLT size. These parameters were also found to have high values of r in our present study. While MPR has the advantage of creating a 3D model of the airway for more precise measurements, it is not practical for emergency patients.

In contrast, the measurements used in our method were easy to acquire; weight, height and chest circumference can be directly measured from the patient, and TD-trachea can be obtained from chest X-ray images. Moreover, a predictive machine learning model trained with local population data could be more tailored to the local population’s average bodily features. The formula derived from machine learning could be easily developed into assisting tools such as mobile device applications. For example, a cell-phone with a dedicated software would allow clinicians to obtain values on ‘depth’ and ‘size’ almost instantly after entering the five parameters.

Some imitations of our study are as follows. First, it had a relatively small sample size. Second, ‘depth’ and ‘size’ were separately modeled instead of using the same model, making the development of future software applications more complicated. Third, we did not test exhaustively all models, and therefore cannot rule out the possibility of other better models than SVM outside linear regression.

## 6. Conclusions

In conclusion, we believe that our predictive model provides clinicians with a more objective reference when it comes to deciding upon DLT size and intubation depth in a clinical setting, emergent or not.

## 7. Patents

We are planning a patent application resulting from this work.

## Figures and Tables

**Figure 1 diagnostics-12-03162-f001:**
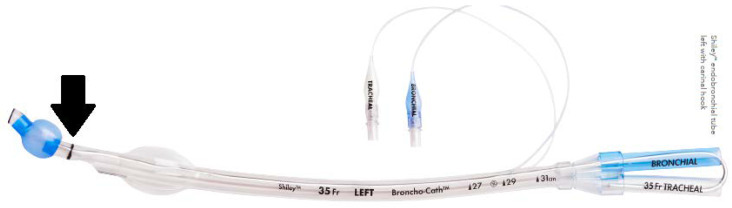
The outer diameter of a left DLT was measured using a caliper at the carina mark (indicated by the black arrow).

**Figure 2 diagnostics-12-03162-f002:**
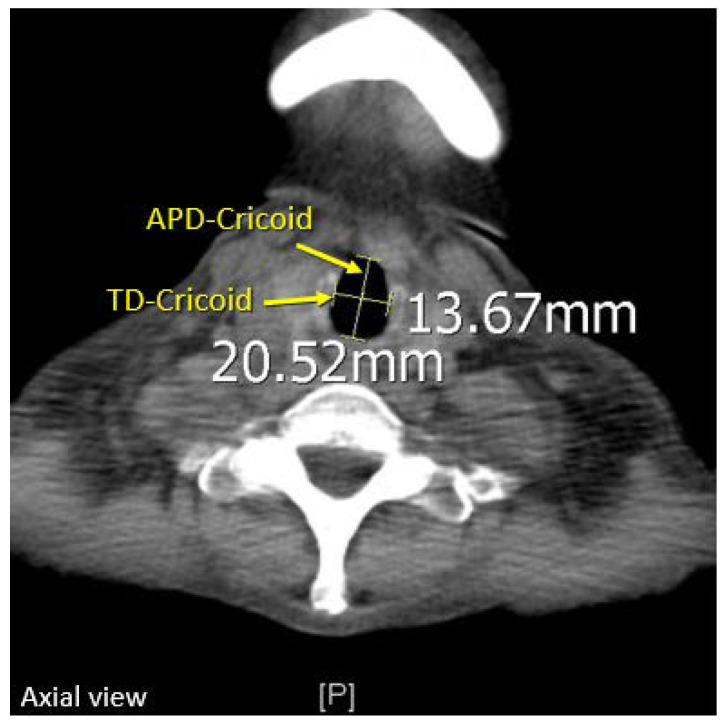
Measuring of the patient’s TD-Cricoid and APD-Cricoid, performed at the lower border of the cricoid cartilage on chest CT.

**Figure 3 diagnostics-12-03162-f003:**
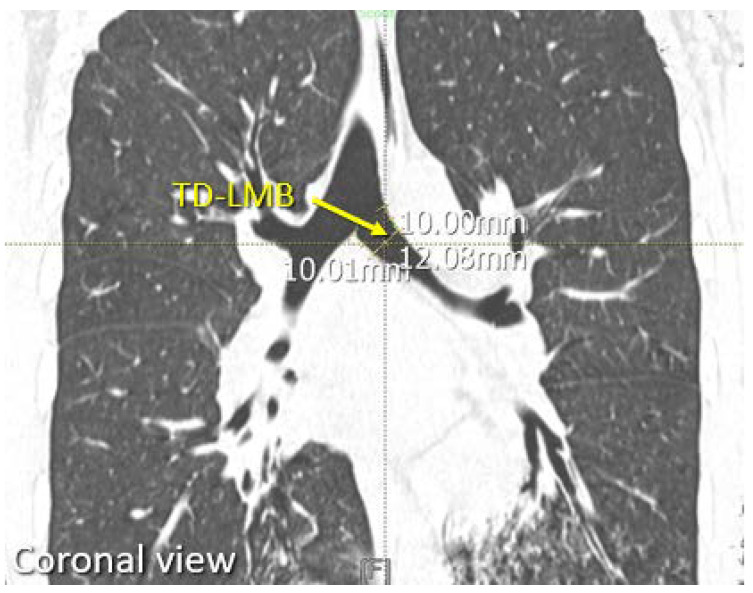
Picture shown measurement of patients’ TD-LMB, performed at 1 cm below carina on chest CT.

**Figure 4 diagnostics-12-03162-f004:**
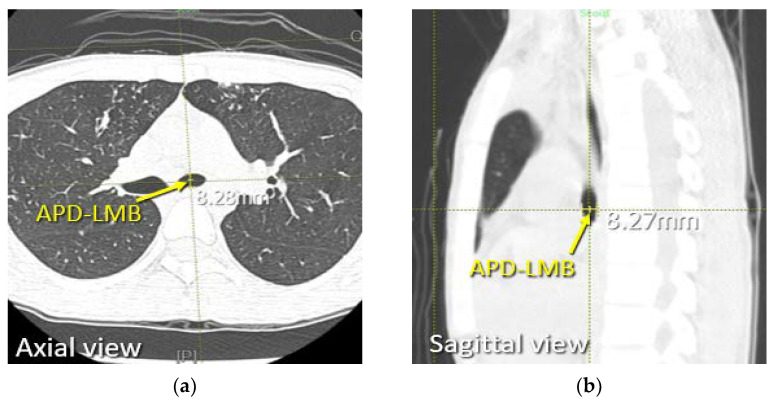
Measuring of the patient’s APD-LMB, performed at 1 cm below carina on chest CT. (**a**) Coronal section of chest CT showing measurement for APD-LMB. (**b**) Sagittal section of chest CT showing measurement for APD-LMB.

**Figure 5 diagnostics-12-03162-f005:**
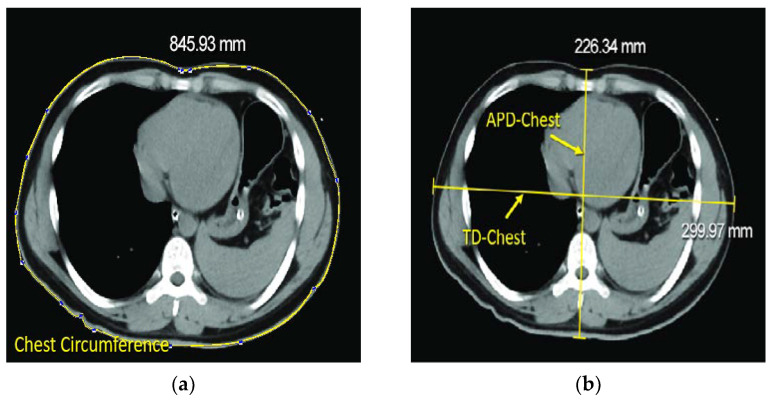
Measuring of the patient’s chest circumference, TD-Chest and APD-chest, performed at the 7th intercoastal space on chest CT. (**a**) Measuring chest circumference by using the built-in measuring tool in PACS. (**b**) TD-Chest and APD-Chest, per-formed at the 7th intercoastal space on chest CT.

**Figure 6 diagnostics-12-03162-f006:**
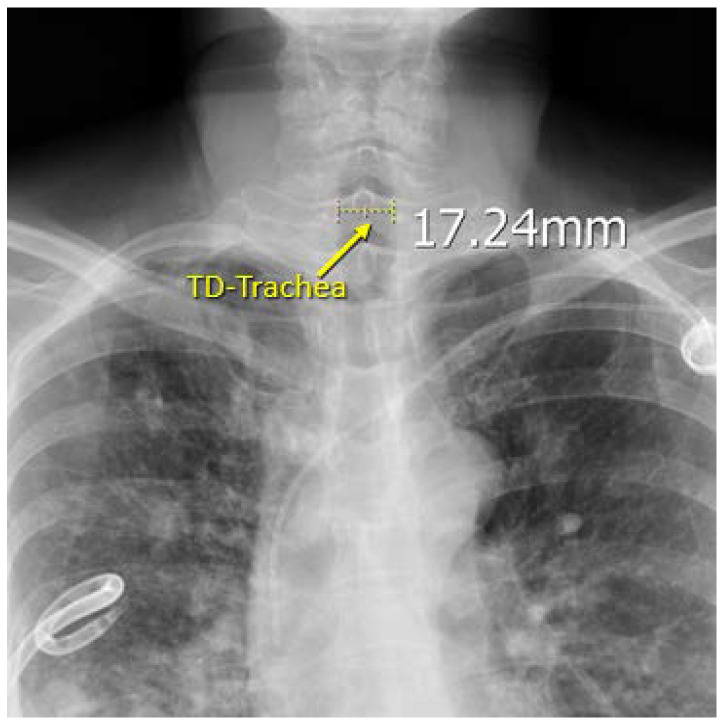
Measuring of the patient’s TD-Trachea, performed at the mid body of C7 on chest X-ray.

**Figure 7 diagnostics-12-03162-f007:**
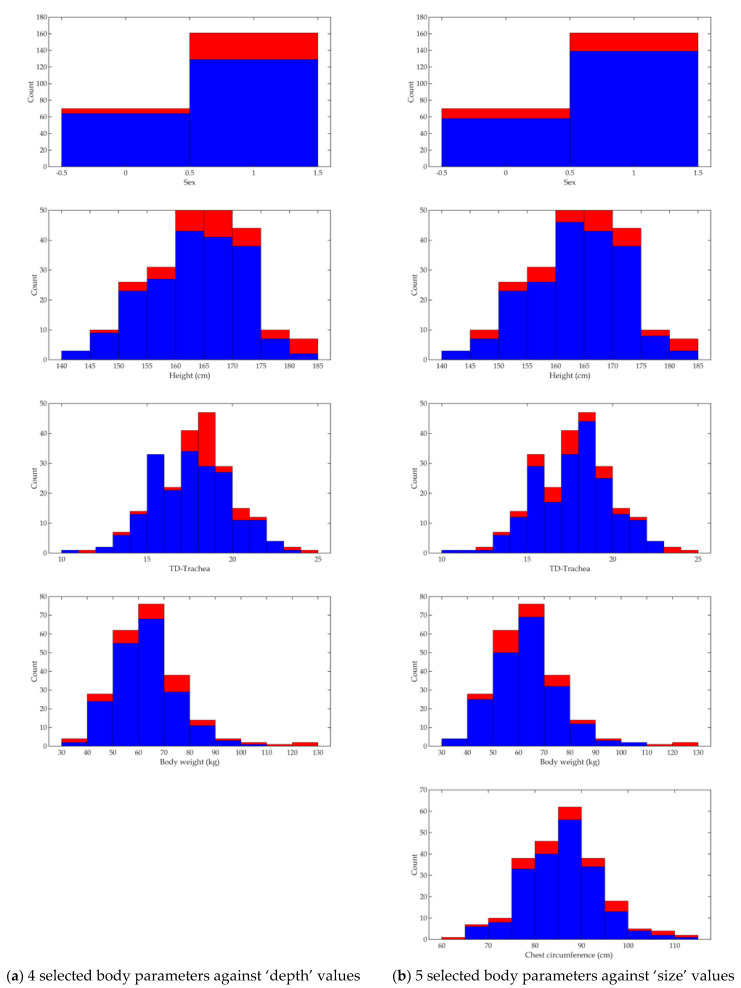
Histograms showing data distributions for four or five selected body parameters against ‘depth’ or ‘size’ values, those after outlier rejection (in blue) and outliers (in red). Note in proportion, more values on both ends of the distributions were rejected.

**Figure 8 diagnostics-12-03162-f008:**
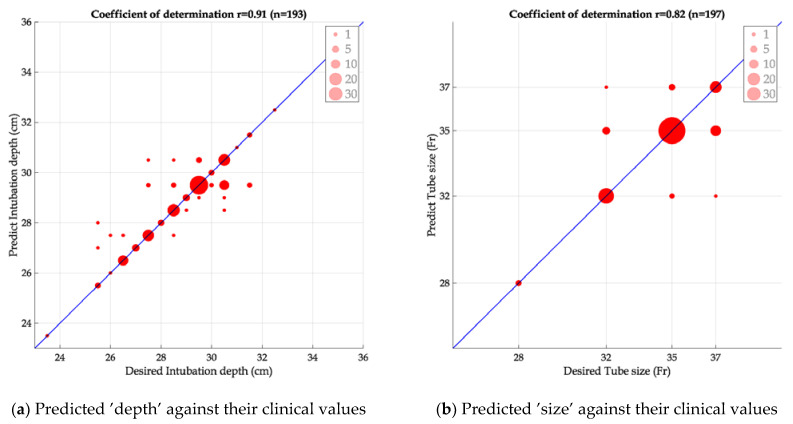
Scatterplots of predicted values of ‘depth’ or ‘size’ against their clinical values. Note that numbers of subjects are different between two panels, as ‘depth’ and ‘size’ were modeled separately.

**Table 1 diagnostics-12-03162-t001:** List of 17 body parameters. Note they were derived from medical records, measured or calculated from X-ray or CT images.

No.	Parameter	Acquire Method
1	SEX	Patient medical record
2	AGE	Patient medical record
3	WEIGHT	Patient medical record
4	HEIGHT	Patient medical record
5	BMI	Calculated (from WEIGHT and HEIGHT)
6	ASA	Patient medical record
7	TD-TRACHEA	Measured from X-ray
8	TD-CRICOID	Measured from CT
9	APD-CRICOID	Measured from CT
10	TD-LMB	Measured from CT
11	APD-LMB	Measured from CT
12	ED-LMB	Calculated (from TD-LMB and APD-LMB)
13	TD-CHEST	Measured from CT
14	APD-CHEST	Measured from CT
15	CHEST-CIRCUMFERENCE	Measured from CT
16	TRACHEA-LMB-ANGLE	Measured from CT
17	RMB-LMB-ANGLE	Measured from CT

**Table 2 diagnostics-12-03162-t002:** Demographic Data. Values are given as number (%) and mean ± standard deviation (minimum–maximum).

Sex	Male: 161 (69.7%)	Female: 70 (30.3%)	Total: 231
**Age (y)**	57 ± 15.8 (14–86)	59.5 ± 14.5 (17–88)	57.7 ± 15.4 (14–88)
**Weight (kg)**	66.2 ± 14.8 (31–130)	57.4 ± 10.2 (38–83)	63.5 ± 14.1 (31–130)
**Height (cm)**	167.1 ± 6.3 (150–185)	154.7 ± 5.4 (143.6–167)	163.4 ± 8.3 (143.6–185)
**BMI**	23.6 ± 4.8 (13.2–40)	24 ± 4.4 (15.4–34.7)	23.8 ± 4.7 (13.2–39.9)

**Table 3 diagnostics-12-03162-t003:** Intubation Data. Values are given as number (%) and mean ± standard deviation (minimum–maximum).

DLT Size	Male	Female	Total
28 Fr	3 (1.9%)	6 (8.6%)	9 (3.9%)
32 Fr	12 (7.5%)	40 (57.1%)	52 (22.5%)
35 Fr	104 (64.6%)	24 (34.3%)	128 (55.4%)
37 Fr	42 (26.1%)	0 (0%)	42 (18.2%)
Intubation Depth (cm)	29.7 ± 1.1 (27–35)	27.2 ± 1.1 (23.5–29.5)	29.0 ± 1.6 (23.5–35)

**Table 4 diagnostics-12-03162-t004:** Roentgenogram Measurements. Values are given as mean ± standard deviation (minimum–maximum).

	Male	Female	Total
**TD-Trachea (mm)**	18.7 ± 1.8 (14.1–24.5)	15.5 ± 1.7 (10.9–20.4)	17.8 ± 2.3 (10.9–24.5)
**TD-Cricoid (mm)**	17.1 ± 1.8 (10.9–22.2)	13.5 ± 1.5 (9.9–16.5)	16.0 ± 2.4 (9.9–22.2)
**APD-Cricoid (mm)**	22.7 ± 2.6 (14–35.2)	18.2 ± 1.7 (14.8–23.2)	21.4 ± 3.1 (14.0–35.2)
**TD-LMB (mm)**	14.5 ± 2 (9.9–22.1)	12.4 ± 1.6 (8.5–19.2)	13.9 ± 2.1 (8.5–22.1)
**APD-LMB (mm)**	12.4 ± 2.1 (6.9–20.2)	10.3 ± 2 (6.9–15.3)	11.8 ± 2.3 (6.9–20.2)
**ED-LMB (mm)**	13.5 ± 1.8 (8.4–20.5)	11.4 ± 1.5 (8.1–15.1)	12.9 ± 2.0 (8.1–20.5)
**TD-Chest (mm)**	320.3 ± 26 (260.4–404.4)	298.5 ± 28.3 (239.2–367.1)	313.7 ± 28.5 (239.2–404.4)
**APD-Chest (mm)**	225.9 ± 30.9 (153.8–336.1)	211.1 ± 27 (138.9–260.3)	221.4 ± 30.5 (138.9–336.1)
**Chest Circumference (mm)**	877.3 ± 80.1 (695.4–1145.2)	817.3 ± 80.1 (634.6–968.3)	859.1 ± 84.6 (634.6–1145.24)
**Trachea-LMB Angle (°)**	140.3 ± 10.5 (115.4–167.1)	139.3 ± 10.8 (113.2–170.5)	140 ± 10.6 (113.2–170.5)
**RMB-LMB Angle (°)**	80.6 ± 15.3 (38.7–115.5)	82.8 ± 17.3 (43.1–123.6)	81.3 ± 16 (38.7–123.6)

**Table 5 diagnostics-12-03162-t005:** Correlation coefficient (r) of 17 individual body parameters against their ’depth’ or ‘size’ values (*n* = 231 subjects), ordered from high r down to low r. The parameters (4 for ‘depth’ and 5 for ‘size’) we identified for modeling are marked in bold letters.

Parameter	Correlation r with ‘Depth’ (FIX-DEPTH)	Parameter	Correlation r with ‘Size’ (TUBE-SIZE)
**SEX**	**0.84**	**SEX**	**0.74**
**HEIGHT**	**0.80**	TD-CRICOID	0.71
TD-CRICOID	0.75	**HEIGHT**	**0.70**
**TD-TRACHEA**	**0.71**	**TD-TRACHEA**	**0.67**
APD-CRICOID	0.68	APD-CRICOID	0.66
TD-LMB	0.63	TD-CHEST	0.65
ED-LMB	0.62	**CHEST-CIRCUMFERENCE**	**0.65**
**WEIGHT**	**0.56**	**WEIGHT**	**0.64**
APD-LMB	0.55	ED-LMB	0.60
TD-CHEST	0.52	TD-LMB	0.59
CHEST-CIRCUMFERENCE	0.50	APD-CHEST	0.56
APD-CHEST	0.43	APD-LMB	0.55
AGE	−0.37	BMI	0.46
RMB-LMB-ANGLE	−0.24	TRACHEA-LMB-ANGLE	0.36
TRACHEA-LMB-ANGLE	0.13	ASA	0.33
ASA	0.12	AGE	0.16
BMI	0.04	RMB-LMB-ANGLE	0.13

**Table 6 diagnostics-12-03162-t006:** Results compared for two models: linear regression, and SVM, with full sets of parameters, or with leave-one-out.

Depth	Linear Regression	Cubic SVM
**4 Parameters**	0.86	0.91
**3 Parameters (no Sex)**	0.79	0.87
**3 Parameters (no Height)**	0.77	0.80
**3 Parameters (no TD-Trachea)**	0.86	0.86
**3 Parameters (no Weight)**	0.85	0.77
**Size**	**Linear Regression**	**Cubic SVM**
**5 Parameters**	0.68	0.82
**4 Parameters (no Sex)**	0.66	0.78
**4 Parameters (no Height)**	0.67	0.68
**4 Parameters (no TD-Trachea)**	0.67	0.75
**4 Parameters (no Chest circumference)**	0.68	0.66
**4 Parameters (no Weight)**	0.67	0.77

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
