# Peer review of "Double-Lumen Endotracheal Tube—Predicting Insertion Depth and Tube Size Based on Patient’s Chest X-ray Image Data and 4 Other Body Parameters"

_diagnostics, 2022, doi:10.3390/diagnostics12123162_

Round 1

Reviewer 1 Report

I totally agree with your conclusion for find an easiest method to measure deep and size of the DLT.

Author Response

Thanks to the comment from reviewer 1, we had sent our manuscript to native English speaker for possible language revision. I hope that the revised manuscript would meet the publication standard of the journal. 

Reviewer 2 Report

The authors have set up the feasible model to predict double lumen tube ‘depths’ and ‘sizes’ to improve clinician’s performance and benefit patients. Very exciting to see it's practice. The only concern is how convenience and how quickly to allow the clinicians to use this model on the emergent circumstance. It’s better to describe more in the limitation.  

Author Response

Q: The only concern is how convenience and how quickly to allow the clinicians to use this model on the emergent circumstance. It’s better to describe more in the limitation.
A: This part on convenience and time is now clarified, first in Results: ‘In addition, the time for SVM modeling was typically fast (<70 msec) for processing data from 231 patients.’; and later in Discussion: ‘For example, a cell-phone with a dedicated software would allow clinicians to obtain values on ‘depth’ and ‘size’ almost instantly after entering the 5 parameters.’

Reviewer 3 Report

1. The manuscript presented is interesting and presents some investigations using endotracheal tube.

2. The abstract seems to lengthy and can be more precise.

3. Organization of the paper is good. But the section numbering need to focused (e.g Section 4.4 is missing.). Similarly, section 4 is repeated and subsequently there off all sections need to be re-numbered.

4. Figure presents a graphical output, but Y-axis representation is missing.

5. Study results are extremely good.

Author Response

Q2. The abstract seems to lengthy and can be more precise.
A2: The abstract is now shortened (from 332 down to 297 words) and it is more precise: ‘In thoracic surgery, the double lumen endotracheal tube (DLT) is used for differential ventilation of the lung. DLT allows lung collapse on the surgical side that requires access to the thoracic and mediastinal areas. DLT placement for a given patient depends on two settings: tube of the correct size (or ‘size’) and to the correct insertion depth (or ‘depth’).
Incorrect DLT placements cause oxygen desaturation or carbon dioxide retention in the patient, with possible surgical failure. No guideline on these settings is currently available for anesthesiologists except the aid from bronchoscopy. Here, we aimed to predict DLT ‘depths’ and ‘sizes’ applied earlier on a group of patients (n=231) using a computer modeling approach. We first determined retrospectively for these patients, the correlation
coefficient (r) of each of 17 body parameters against ‘depth’ and ‘size’. Those parameters having r >0.5 and could be easily obtained or measured were selected. They were, for both DLT settings: (a) sex, (b) height, (c) tracheal diameter (measured from X-ray), and (d) weight. For ‘size’ a fifth parameter, (e) chest circumference was added. Based on these 4 or 5 parameters we modeled the clinical DLT settings using a Support Vector Machine (SVM). After excluding statistical outliers (± 2 SD), 83.5% subjects were left for ‘depth’ in its modeling, and similarly 85.3% for ‘size’. SVM predicted ‘depths’ matched with their clinical values at a r of 0.91, and for ‘sizes’, at a r of 0.82. The less satisfactory result on ‘size’ prediction was likely due to the small target choices (n=4) and the uneven data distribution. Furthermore, SVM outperformed other common models like linear regression.
     In conclusion, this first model on predicting the two DLT key settings gave satisfactory results. Findings would help anesthesiologists in applying DLT procedures more confidently in an evidence-based way.’
Q3. But the section numbering need to focused (e.g Section 4.4 is missing.). Similarly, section 4 is repeated and subsequently there off all sections need to be re-numbered. 
A3: This part is now corrected.
Q4. Figure presents a graphical output, but Y-axis representation is missing.
A4: This part is now corrected.